# CXCR4 Expressed by Tumor-Infiltrating B Cells in Gastric Cancer Related to Survival in the Tumor Microenvironment: An Analysis Combining Single-Cell RNA Sequencing with Bulk RNA Sequencing

**DOI:** 10.3390/ijms241612890

**Published:** 2023-08-17

**Authors:** Chen Su, Rong Yu, Xiaoquan Hong, Panpan Zhang, Yingying Guo, Jian-Chun Cai, Jingjing Hou

**Affiliations:** 1Department of Gastrointestinal Surgery, Zhongshan Hospital of Xiamen University, School of Medicine, Xiamen University, Xiamen 361102, China; suchen@xmu.edu.cn (C.S.); yrong0502@163.com (R.Y.); 2Institute of Gastrointestinal Oncology, School of Medicine, Xiamen University, Xiamen 361102, China; 24520210157044@stu.xmu.edu.cn (X.H.); 21620160154361@stu.xmu.edu.cn (P.Z.); 21620150150559@stu.xmu.edu.cn (Y.G.); 3Department of General Surgery, Zhongshan Hospital of Xiamen University, School of Medicine, Xiamen University, Xiamen 361102, China; 4State Key Laboratory of Stress Cell Biology, School of Life Sciences, Xiamen University, Xiamen 361102, China

**Keywords:** single-cell RNA sequencing, *CXCR4*, bulk RNA sequencing, tumor-infiltrating B cells, gastric cancer

## Abstract

According to the World Health Organization (WHO), gastric cancer (GC) is the fourth leading cause of tumor-related mortality globally and one of the most prevalent malignant tumors. To better understand the role of tumor-infiltrating B cells (TIBs) in GC, this work used single-cell RNA sequencing (scRNA-Seq) and bulk RNA sequencing (bulk RNA-Seq) data to identify candidate hub genes. Both scRNA-Seq and bulk RNA-Seq data for stomach adenocarcinoma (STAD) were obtained from the GEO and TCGA databases, respectively. Using scRNA-seq data, the FindNeighbors and FindClusters tools were used to group the cells into distinct groups. Immune cell clusters were sought in the massive RNA-seq expression matrix using the single-sample gene set enrichment analysis (ssGSEA). The expression profiles were used in Weighted Gene Coexpression Network Analysis (WGCNA) to build TCGA’s gene coexpression networks. Next, univariate Cox regression, LASSO regression, and Kaplan–Meier analyses were used to identify hub genes in scRNA-seq data from sequential B-cell analyses. Finally, we examined the correlation between the hub genes and TIBs utilizing the TISIDB database. We confirmed the immune-related markers in clinical validation samples using reverse transcriptase polymerase chain reaction (RT-PCR) and immunohistochemistry (IHC). 15 cell clusters were classified in the scRNA-seq database. According to the WGCNA findings, the green module is most associated with cancer and B cells. The intersection of 12 genes in two separate datasets (scRNA and bulk) was attained for further analysis. However, survival studies revealed that increased *C-X-C motif chemokine receptor 4* (*CXCR4*) expression was linked to worse overall survival. *CXCR4* expression is correlated with active, immature, and memory B cells in STAD were identified. Finally, RT-PCR and IHC assays verified that in GC, *CXCR4* is overexpressed, and its expression level correlates with TIBs. We used scRNA-Seq and bulk RNA-Seq to study STAD’s cellular composition. We found that *CXCR4* is highly expressed by TIBs in GC, suggesting that it may serve as a hub gene for these cells and a starting point for future research into the molecular mechanisms by which these immune cells gain access to tumors and potentially identify therapeutic targets.

## 1. Introduction

Based on the World Health Organization (WHO) (https://gco.iarc.fr/ accessed on 11 December 2020), gastric cancer (GC) is the fourth leading cause of tumor-related mortality globally and one of the most prevalent malignant tumors. Despite advancements in treatment, such as molecularly targeted therapy, chemotherapy, and surgery, GC outcomes remain dismal [1,2,3,4].

Cancer, stromal, and immune cells coexist in the intricate ecology known as the tumor microenvironment (TME), including immunological infiltrates. Cancer progression can be affected by cross-talking among these cell types. Earlier research suggested that tumor-infiltrating immune cells (TIICs) are key in the initiation, progression, and development of cancer and that the responses of innate and adaptive immune cells determine the success of immunotherapies [5,6,7,8]. Humoral immunological responses rely heavily on the adaptive immune system, particularly B and T cells [9,10]. Antitumor immunity relies on B cells for several reasons, including antibody generation, cytokine release, antigen presentation, and the development of lymphoid architecture. Recent research has shown the crucial function of B cells in cancer immunotherapies [11,12,13]. By secreting soluble mediators to modulate the proangiogenic and protumorigenic actions of myeloid cells or by generating molecules that assist signal transduction in cancer cells, tumor-infiltrating B cells (TIBs) may promote tumor growth by blocking T-cell-mediated immune responses [14,15,16].

The technology for single-cell RNA sequencing (scRNA-seq) is now being utilized to profile cell populations in tumors at the single-cell level and to evaluate the differential molecular characteristic across various types of cell compositions. scRNA-seq allows for a more detailed examination of transcriptome expression patterns at the level of the individual cell [17,18]. This work identified essential genes linked with TIBs in GC through scRNA-seq and RNA-seq studies on human stomach adenocarcinoma (STAD) B cells.

## 2. Results

### 2.1. Examining Immunological Cells within scRNA-Seq Samples

The flow chart of this study was shown in Figure 1. After quality control, a total of 27,547 genes in 16,017 cells were obtained from the scRNA-seq data. The number of genes (nFeature), the sequence count per cell (nCount), and the percentage of mitochondrial genes (percent. mt) were displayed in violin plots (Figure 2D). Low proportions indicate good-quality cells, as the mitochondria are larger than individual transcript molecules and less likely to escape through tears in the cell membrane [19]. The data were log-normalized, and 15 clusters were generated via dimensionality reduction. To examine the cell markers in all 15 groups, we used the “FindAllMarkers” tool (logFC = 0.5, Minpct = 0.25). (Figure 2A,B). Marker genes were used for the detected groups. B cells, dendritic cells, monocytes, neutrophils, and T cells were identified using cell-type annotation (Figure 2C). FC in relative gene expression was determined previously, and |logFC| ≥ 1 and a *p*-value of < 0.05 were used as cutoffs for selecting genes for further analysis. These genes for B cells were found to be uniquely expressed. Figure 2E displays the Gene Ontology (GO) analysis results, which show that the marker genes of the B cells from these pathways, such as cytoplasmic translation, response to an unfolded protein and topologically incorrect protein, positive regulation of leukocyte activation and cell activation, etc. Cluster analysis of B cells, dendritic cells, monocytes, neutrophils, and T cells was based on immune cell markers (Figure 2F).

### 2.2. Infiltrating Immune Cell Count

The invasion of immune cells was measured quantitatively with mRNA data using the single-sample gene set enrichment analysis (ssGSEA). In the end, we were able to collect 28 invading immune cells, which included type 17 and type 1 T-helper cells; activated CD4 T, dendritic, and CD8 T cells; central memory CD4 and CD8 T cells; immature B cells; CD56 bright natural killer cell; effector memory CD4 and CD8 T cells; natural killer T cells; macrophages; immature dendritic cells; CD56 dim natural killer cells; neutrophils; plasmacytoid dendritic cells; type 2 T-helper cells; MDSC; regulatory T cells; eosinophils; gamma delta T cells; activated B cells; mast cells; follicular helper cells; monocytes; memory B cells. Z-scores for each sample and invading immune cell pair (Figure 3), as well as for three kinds of B cells, were calculated for Weighted Gene Coexpression Network Analysis (WGCNA) using this algorithm.

### 2.3. The WGCNA Structure and Identification of Essential Modules

WGCNA was used to examine the preprocessed data and identify the modules of highly linked genes. The 14,857 gene expression patterns were analyzed using WGCNA after removing five outliers. To guarantee a scale-free network, we used a soft threshold of = 6 (scale-free R^2^ = 0.88, Figure 4A,B). We constructed the network and determined the soft threshold simultaneously. The module’s minimal gene count was set to 30. To obtain five modules, we began by dynamically shearing a tree with an abline of 0.9 to create two new modules from the original. Then, we combined the modules containing feature genes that are very comparable using the gene cluster dendrogram (Figure 4C). We used a heat map to examine the relationship between cancer, B cells, and the module’s genes. The green module showed the strongest connection with B-cell activity, B-cell immaturity, memory B cells, and cancer. Thus, the top 2000 genes in the green module were chosen for further investigation because of their strong link with cancer and other types of B cells. Zhang Bin et al. provide a comprehensive description of the WGCNA algorithm in [20].

### 2.4. Analysis of the Relationship between Key Genes and Prognosis in Gastric Cancer

We identified an intersection between the scRNA and WGCNA dataset. Figure 5A,B show the B-cell marker genes that were identified as being uniquely expressed. Using univariate Cox regression analysis, we found genes associated with prognosis: those with hazard ratios >1 are associated with a poor prognosis, while those with hazard ratios <1 are protective. Among the 12 genes identified before, the genes *CXCR4*, *TSC22D3*, *YPEL5*, *GADD45B*, and *ZNF331*, which had an HR > 1, were independent risk factors (Figure 6A). These five important genes were then analyzed independently for their effects on survival.

### 2.5. Screening the Key Genes with LASSO

For the further screening of those five genes, we turned to LASSO. The adjustment parameter λ was estimated through cross-validation (Figure 6B,C). *CXCR4* and *GADD45B* were the two most important genes to focus on.

### 2.6. GEPIA Database Analysis

Using the GEPIA website, we correlated *CXCR4*, *TSC22D3*, *GADD45B*, and *ZNF331* expression levels with the Overall Survival (OS) rate of patients with STAD. We found high levels of *CXCR4*, *TSC22D3*, *GADD45B*, and *ZNF331* expression were related to worse overall survival in STAD (Figure 6D).

### 2.7. TISIDB Database Verification of the Association between Key Genes and Immune Cell Invasion

We obtained the intersection of key genes in five separate datasets (scRNA, WGCNA, uniCox, LASSO, and Kaplan–Meier) (Figure 6E). *CXCR4* and *GADD45B* contained in the intersection were identified as the hub gene, and we chose *CXCR4* for further analysis. We aimed to further verify the association between the hub gene expression of *CXCR4* and other B-cell infiltration types in STAD. Using the TISIDB database, the expression of the hub gene was evaluated. Figure 7A demonstrates a substantial positive correlation between *CXCR4* expression and active, immature, and memory B cells. Based on our findings, the infiltration of active, immature, and memory B cells may be a significant source of *CXCR4* expression in the milieu surrounding STAD.

### 2.8. Gene Set Enrichment Analysis

It was discovered that the biological processes most enriched in the *CXCR4* dataset were strongly connected with immune responses, including cell recognition (NES = 2.4839008), complement activation (NES = 2.483554), leukocyte proliferation (NES = 2.4473825), negative regulation of immune system processes (NES = 2.431409), natural killer cell activation (NES = 2.409364), positive regulation of chemotaxis (NES = 2.4061759), etc. (Figure 7B). These results prove that *CXCR4* is related to immune responses.

### 2.9. CXCR4 Is Overexpressed in STAD

First, we employed RT-PCR to evaluate the *CXCR4* mRNA level in 30 pairs of GC cells. In sixty percent of human GC tissues, *CXCR4* levels were significantly elevated compared to normal tissue levels (22 out of 30) (Figure 8). Table 1 further shows a statistically significant correlation between *CXCR4* levels and TNM staging (*n* = 30, *p* < 0.05), but there was no clear correlation between *CXCR4* expression and age, gender, or tumor size.

### 2.10. CXCR4 Expression Correlates Positively with TIBs in Clinical Samples of STAD

To investigate the link between CXCR4 and tumor-infiltrating B lymphocytes in GC, we performed immunohistochemical analysis with a GC tissue chip containing 80 pairs of clinical samples. CD19 and CD20 were chosen and examined as an indication of tumor-infiltrating B lymphocytes; the immunohistochemical score was determined by multiplying the staining intensity score by the staining extent score. The color depth of brown is positively correlated with the protein expression level. The results indicate that in roughly 70–75% of human GC tissues, CD19 and CD20 were highly upregulated compared to the controls, which was compatible with the expression of CXCR4 (Figure 9A). Significant positive associations were found between the expression of CXCR4 and the expression of CD19 and CD20 proteins, according to correlation studies (Figure 9B). Our findings indicate that the upregulation of CXCR4 may influence the microenvironment of GC by influencing the immunological responses of TIBs. CXCR4 may serve as a target for immunotherapy.

## 3. Discussion

The presence or absence of immune infiltrates in the TME influences clinical outcomes for cancer patients. TIBs may be found in at least two separate TME structural zones: highly structured tertiary lymphoid structures (TLSs), which are similar to lymph nodes; and a less structured stromal infiltration with fewer macrophages, T cells, and TIBs that we term lymphoid–myeloid aggregates (LMAs) [21]. TIBs exist in immunologically “hot” tumors associated with myeloid cells, T cells, and others and have intensive predictive and prognostic value in immune checkpoint blockades and tumor treatments. Inhibiting self-tolerance pathways is one way TIBs may fight immune editing and tumor heterogeneity; they help set up and maintain tumor microenvironments rife with immune cells such as natural killer cells, myeloid cells, and T cells, and they present antigens to T cells in a novel way. Germinal center B cells; naive, memory, and activated B cells; plasma cells; and their intermediates were all identified in the first investigations of TIBs morphologies in human cancer [22]. Here, we utilized the Seurat package in R to uncover the inverse correlation between B cells and marker genes, supported by the data from scRNA-Seq databases. These marker genes showed enrichment in several GO categories, including “response to unfolded protein”, “reaction to topologically wrong protein”, “positive regulation of cell activation”, and “positive regulation of leukocyte activation”.

According to the ssGSEA, activity B cells, memory B cells, and immature B cells were included in our study. Considering the above situation, we identified a significant transcriptional induction of the genes most relevant to these types of B cells by combining small-scale RNA-seq and large-scale RNA-seq. We obtained 15 cell clusters from the scRNA-seq profiles of 27,547 genes across 16,017 cells from three STAD patients. Furthermore, these gene expression levels and the patients’ OS rate were also investigated with STAD. Univariate Cox regression analysis revealed that high abundances of *CXCR4*, *TSC22D3*, *YPEL5*, *GADD45B*, and *ZNF331* were associated with worse survival. According to the WGCNA findings, the green module is most associated with malignancy and the B-cell cluster. We merged five datasets to find common groups of genes (scRNA, WGCNA, uniCox, LASSO, and Kaplan–Meier). *CXCR4* and *GADD45B* were the central genes in the junction. *CXCR4* was chosen for further analysis.

Breast cancer metastasis to the lungs was the first tissue in which the *CXCR4*, a 352-amino-acid rhodopsin-like GPCR, was identified [23]. *CXCR4* is the most abundantly expressed chemokine receptor in more than 23 human cancers, including breast, ovarian, melanoma, prostate, and colorectal cancer, while being expressed at low or undetectable levels in numerous normal tissues [24,25,26]. Malignant ascites and peritoneal carcinomatosis are two symptoms of primary gastric carcinomas that may be traced back to the *CXCL12-CXCR4* axis [27]. Tumor stage, invasion depth, lymph node metastases, vascular invasion, and poor prognosis were strongly linked with *CXCR4* expression in patients with GC [28,29]. This study’s examination of the TISIDB database indicates a strong positive correlation between *CXCR4* expression and the three types of B cells (active, immature, and memory). Our findings suggest a link between *CXCR4* and B cells in STAD, suggesting *CXCR4* might be a valuable therapeutic target for this disease. *CXCR4* signaling has been proposed as an acquired resistance mechanism in BCR-dependent diffuse large B-cell lymphomas. Prior research has classified *CXCR4* overexpression as an indication of susceptibility to *BCR/PI3K* blocking [30]. By examining enriched gene sets, we found that immunological responses were highly correlated with *CXCR4* in the dataset we analyzed. Lastly, we demonstrated that *CXCR4* expression levels were high in STAD tissues and correlated with B cells using IHC and RT-PCR.

## 4. Materials and Methods

### 4.1. Expression Profile Information on Genes

STAD scRNA-seq data from GSE163558 and three GC patients’ records, along with other data, were retrieved from GEO. The TCGA provided RNA-seq data at STAD level 3 for 32 normal and 375 primary tumor tissues. Also, clinical data were collected.

### 4.2. Clustering Dimension Reduction for scRNA Seq Data and Immune Cell Screening

Initially, we used the log-normalization function to normalize the merged data and then used the FindVariableFeatures tool to identify the top 2000 highly variable genes (variable features were identified using variance stabilization transformation (“vst”)). Simultaneously, ScaleData was used to scale all gene values, and RunPCA was executed to reduce the PCA dimension for the top 2000 most variable genes. To locate the cell clusters, we set dim = 30 and clustered the cells using “FindNeighbors” and “FindClusters” (resolution = 0.5). Subsequently, we utilized the ScaleData function to refine the data by shifting the gene expression of each cell to yield an average expression of 0. Additionally, we scaled the gene expression of each cell to ensure that the variation between cells was equal to 1. Once the data were normalized, the RunPCA and DimPlot routines were used to illustrate PCA for the 2000 highly variable genes. The elbow plot technique was used to obtain the total number of PCs. In principal component analysis, we used Euclidean distance to build a KNN graph. The edge weights between any two cells were determined using the FindNeighbors function. The FindClusters method was used to examine cell clusters. Methods for reducing dimensions not similar to those of linear cells were grouped in a 2-dimensional space using t-distributed statistical neighbor embedding (tSNE). The “Seurat” R package’s “FindAllMarkers” function was used to screen the marker genes in each group. Using the “SingleR Version 2.2.0” software and the Monaco database, we next determined the cell types and linked the marker genes from each cluster to the appropriate cell types. Ultimately, we screened the marker gene using a corrected *p*-value of <0.05 and a logFC value of >1. Marker genes of B cells were clustered for Gene Ontology (GO) enrichment analysis using the clusterProfiler package [31].

### 4.3. Infiltration of Immune Cells

We used the GSVA package to compare the gene expression profile data from TCGA and the immune cell metagenes set, enabling us to calculate the enrichment scores for every immune-related phrase [32]. By utilizing this methodology, we identified 28 distinct types of immune cells within the tumor microenvironment. The unique gene panels of each immune cell were extracted from a recent publication, enabling us to accurately detect their presence [33]. The enrichment score was determined by adding the variances between the empirical cumulative distribution functions of gene rankings. The ssGSEA algorithm assigns rankings to genes based on their absolute expression levels within a given sample. We standardized the ssGSEA score to ensure that each kind of immune cell is represented by a score ranging from 0 (lowest possible score) to 1 (highest possible score).

### 4.4. Weighted Gene Coexpression Network Analysis (WGCNA)

A gene coexpression network was built using the WGCNA R package [34]. We began by selecting the top quartile of gene expression levels, which accounts for around a quarter of the total variation in gene expression. After calculating standard deviations and gathering pertinent data, the top 14,857 changing genes in the TCGA dataset were identified, representing the 407 STAD samples used. Outlier samples with connection values below 2.5 were removed before the clustering tree map was created [35]. A sample tree diagram and a trait heat map to illustrate genes for orthologous were created. The spectrum may be broken down further into its component gene modules. After removing 25 outlier samples, these integrated genes’ gene expression profiles and relevant information were used as input datasets for WGCNA to build a representative dendrogram and trait heat map. The R2 index for scaling-free topology fitting was computed using the pickSoftThreshold function for various values of the soft-thresholding power β. The R2 of 0.8 was used as the criterion to select values. In light of this, β = 3 was chosen as the soft threshold in this investigation. After the sensitivity threshold was established, the network could be constructed. It is possible to construct a network in which the gene dendrogram and nodule color are determined by the degree of dissimilarity using a Topological Overlap Matrix (TOM) produced from the gene expression matrix. A dissTOM-based hierarchical clustering tree (dendrogram) of gene functions was generated using hierarchical clustering for module discovery. The module’s minimal gene count was set to 30. The original module was halved using dynamic tree shearing, abline = 0.25 was applied, and the modules that shared the most similar feature genes were merged to produce 13 modules based on the gene dendrogram.

Cancer, B-cell activity, B-cell maturity, and memory B cells were all considered. We calculated the Pearson correlation coefficient between a sample vector of these variables and the module’s signature gene to assess the robustness of the link between the clinical aspects and the module. This allowed us to create a correlation analysis diagram that displays the relationship between the clinical data and the gene module to find out which gene modules in immune cells are crucial for different B cells. In addition, we needed to ascertain the degree of interconnectivity inside the module and choose the top 2000 connected genes for further studies.

### 4.5. Analysis for Univariate Cox Regression

The “survival” package in R was used for the univariate Cox regression analysis. Hazard ratios and appropriate confidence intervals were estimated using the univariate Cox model to assess the association between the major genes involved in prognosis. *p* < 0.05 was considered statistically significant.

### 4.6. Screening Key Genes via LASSO

Target genes were identified through univariate analysis, and then LASSO regression was used to produce model improvement by generating a penalty function to restrict the model size and avoid overfitting. 

### 4.7. Prognosis Database Analysis

Using the GEPIA (Gene Expression Profiling Interactive Analysis) website (http://gepia.cancer-pku.cn/index.html accessed on 3 March 2023), we looked at the association between the expression levels of these key genes and the OS rate of patients with STAD.

### 4.8. TISIDB Database Verification of the Association between Essential Genes and Immune Cell Infiltration

TISIDB is an online site (http://cis.hku.hk/TISIDB/index.php accessed on 3 March 2023) for tumor and immune system interaction that incorporates many forms of heterogeneous data and examines correlations between target genes and lymphocytes [36]. In this work, using STAD, we wanted to identify the levels of activity, memory, and immature B cells and the connection between the expression of important genes and these subsets. We analyzed STAD expression scatter plots for statistical significance using Spearman’s correlation.

### 4.9. Gene Set Enrichment Analysis

Java GSEA (version 4.3.2) was used, and the gene set “c5.go.bp.v2023.1.Hs.symbols.gmt” from the GO database was selected as the reference [37]. Biological processes with a normalized *p*-value below 0.05 and a false discovery rate (FDR) q-value below 0.05 were deemed statistically significant. The leading changed biological processes were selected based on a ranking of normalized enrichment scores (NESs).

### 4.10. Clinical Samples 

The Institutional Medical Ethics Committee of Xiamen University authorized all research methods, and all clinical samples were collected with the patients’ informed permission in compliance with the principles outlined in the Declaration of Helsinki (1975). At least two specialized pathologists confirmed the GC pathological diagnosis. Eighty human GC specimens and their corresponding surrounding epithelial tissues were collected from Shanghai, China, Outdo Biotech Co., Ltd. 

### 4.11. Extraction of RNA via RT-PCR

The cDNA chip containing 30 pairs of GC tissue was purchased from Shanghai Outdo Biotech Co., Ltd., Zhangjiang Science City, China. RT-PCR was used to amplify and evaluate the generated cDNA samples using Power SYBR Green PCR Master Mix (Applied Biosystems, Foster City, CA, USA) and an ABI 7500 Real-Time PCR System (Applied Biosystems, Foster City, CA, USA). The primers used in quantitative real-time polymerase chain reactions are as follows: human CXCR4, 5′-ACGCCACCAACAGTCAGA-3′ (forward) and 5′-CACAACCACCCACAAGTCA-3′ (reverse); human GAPDH, 5′-GACATCAAGAAGGTGGTGAA-3′ (forward) and 5′-TGTCATACCAGGAAATGAGC-3′ (reverse).

### 4.12. IHC Staining

Through Outdo BioTech, we acquired one human GC TMA (Shanghai, China). Moreover, our prior work described the specific experimental methods and immunohistochemistry scores used to evaluate the samples [3]. Briefly, gastric cancer and paracancerous tissues were fixed in 10% formalin, paraffin-embedded, sliced into 4-6 um sections, and placed onto slides. After deparaffinization, rehydration, and microwave antigen retrieval, the slides were incubated with primary antibodies against CD20 (1:5000 dilution; Proteintech Group, Inc., Rosemont, IL, USA), CD19 (1:5000 dilution; Proteintech Group, Inc., Rosemont, IL, USA), and CXCR4 (1:1000 dilution; ab214050; Abcam, Cambridge, MA, USA) antibodies at 4 °C overnight. Afterward, the slides were incubated with a secondary antibody at room temperature for 30 min and stained with DAB substrate, followed by hematoxylin counterstaining. The immunohistochemical score was determined by multiplying the staining intensity score by the staining extent score. The positive result is a brown color of varying shades. The color depth of brown was positively correlated with the protein expression level.

### 4.13. Analyses of Statistics

Statistical tests on GEO and TCGA were performed in R 4.2.2. (R Institute for Statistical Computing, Vienna, Austria). Above, we detailed the R packages used for these statistical studies. The statistical tests with RT-PCR and IHC were performed using GraphPad Prism 9.0 (GraphPad Software, Inc., San Diego, CA, USA), and the correlation analyses proceeded with Pearson’s test. Comparisons between groups were made via Fisher’s exact analysis using IBM SPSS Statistics version 27 (IBM Corp, Armonk, NY, USA). When the *p*-value was less than 0.05, it was regarded as significant. 

## 5. Conclusions

In conclusion, by using bulk RNA-seq and scRNA-seq data and conducting WGCNA, we identified the potential for *CXCR4* to act as a novel target for STAD therapy, which led us to investigate its function in this disease.

## Figures and Tables

**Figure 1 ijms-24-12890-f001:**
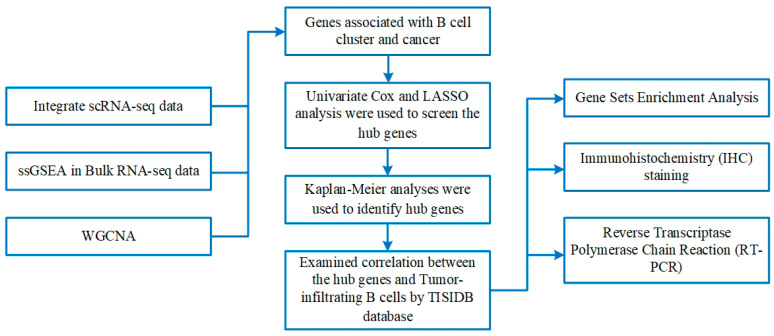
The flowchart of this investigation.

**Figure 2 ijms-24-12890-f002:**
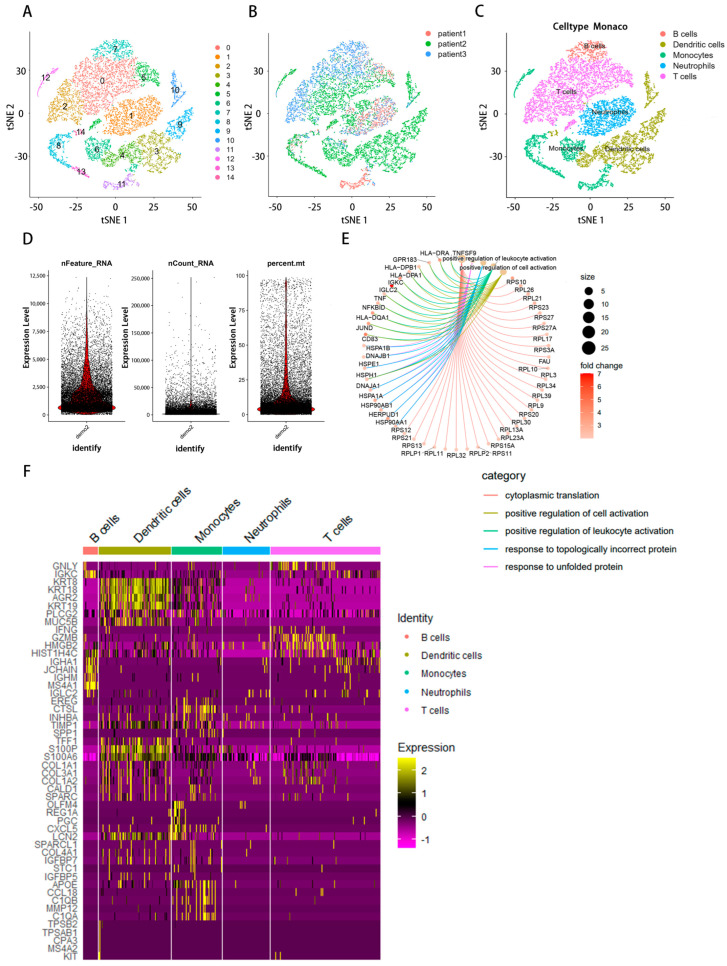
The dimension reduction in STAD scRNA-seq. (**A**) Colors are labeled based on distinct groups. (**B**) Different colors indicate different patients. (**C**) Different colors indicate different immune cells. (**D**) Quality control of scRNA-seq data of samples of STAD cells, the number of genes (nFeature), the sequence count per cell (nCount), and the percentage of mitochondrial genes (percent. mt) in scRNA-seq are displayed in violin plots. (**E**) The cnetplot presented the network of maker genes of B cells from these pathways. Colored points referred to the corresponding pathways. (**F**) Heat map of gene expression in immune cells.

**Figure 3 ijms-24-12890-f003:**
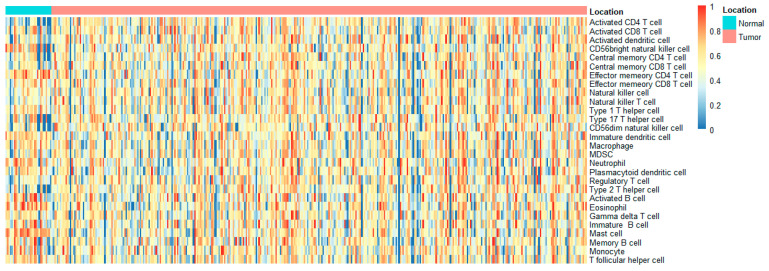
An analysis of gene set enrichment using single samples identified the relative recruitment of immune cells in STAD. The heat map reveals the z-scores calculated based on each cell type’s relative infiltration.

**Figure 4 ijms-24-12890-f004:**
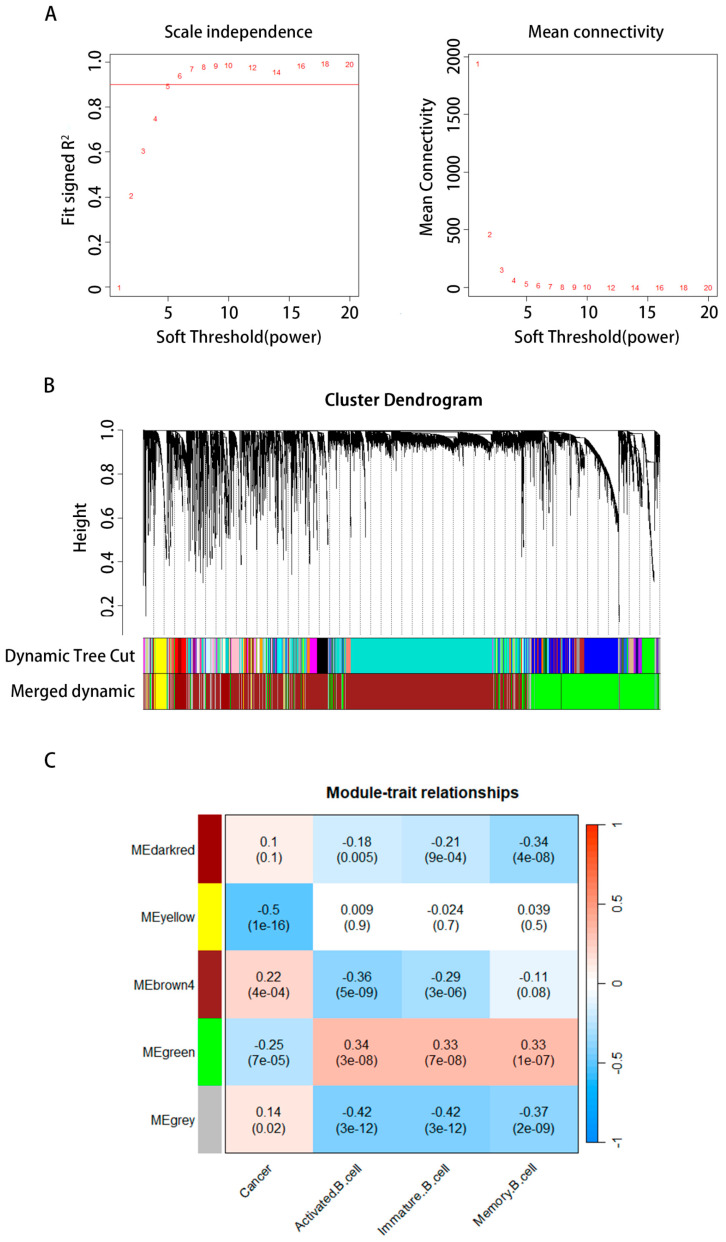
Screening for hub genes and immunotherapy prediction. (**A**) The characteristics of the network architecture were established using distinct power values; the correlation between power and average connection is shown. (**B**) The clustering of genes into distinct groups. (**C**) Correlation examination of modules with trait modules, with the green module being the most significant module for cancer and the number of various types of B cells.

**Figure 5 ijms-24-12890-f005:**
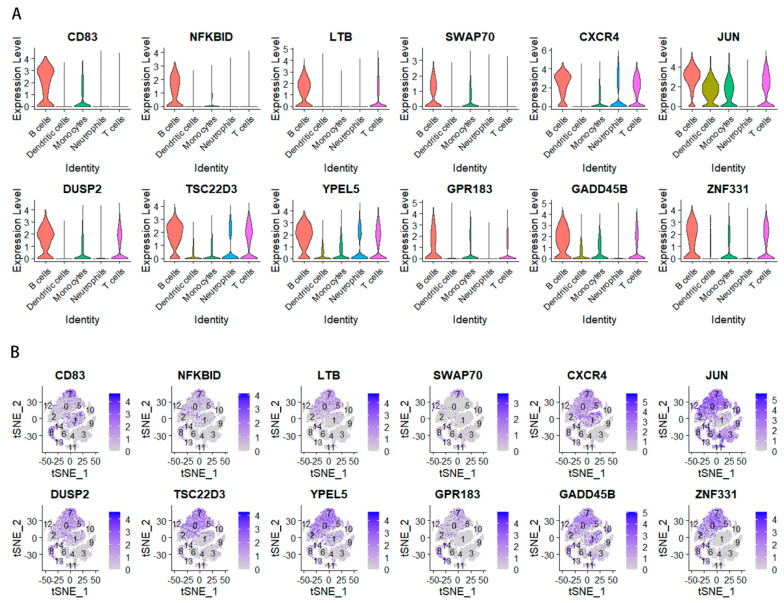
(**A**) A violin diagram of the 12 genes. (**B**) Expression of important marker genes in the dimension reduction in STAD scRNA-seq. Color scale indicates expression level.

**Figure 6 ijms-24-12890-f006:**
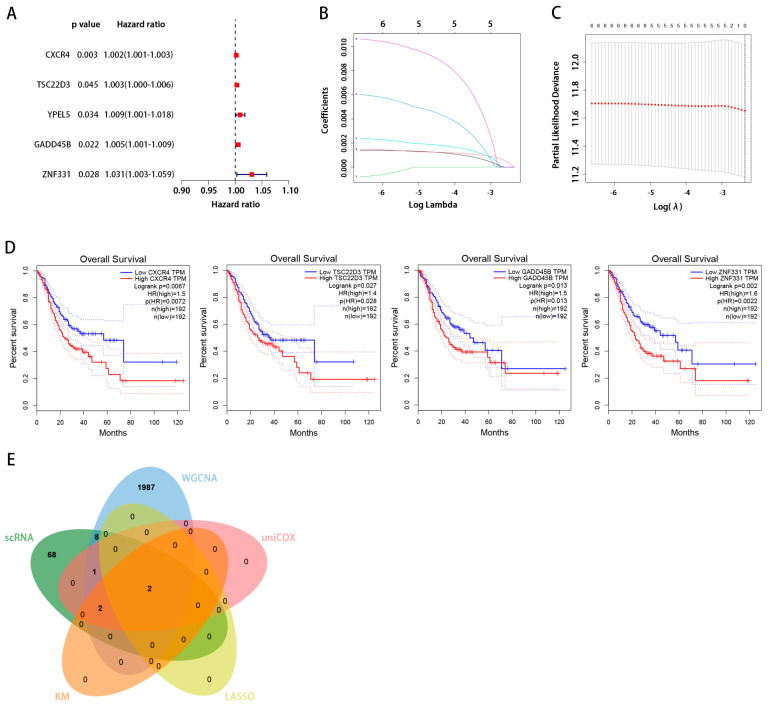
(**A**) The univariate analysis of marker genes indicated that *CXCR4*, *TSC22D3*, *YPEL5*, *GADD45B*, and *ZNF331* are poor prognostic factors. (**B**) Five genes were chosen to construct a risk model. The trajectory of each independent variable is as follows: the horizontal axis represents the log value of the independent variable lambda, and the vertical axis represents the coefficient of the independent variable. (**C**) The confidence interval under each lambda. (**D**) The Kaplan–Meier survival curves indicated that *CXCR4*, *TSC22D3*, *GADD45B*, and *ZNF331* had a considerably poorer prognosis. (**E**) Acquisition of key genes associated with tumor-infiltrating B cells.

**Figure 7 ijms-24-12890-f007:**
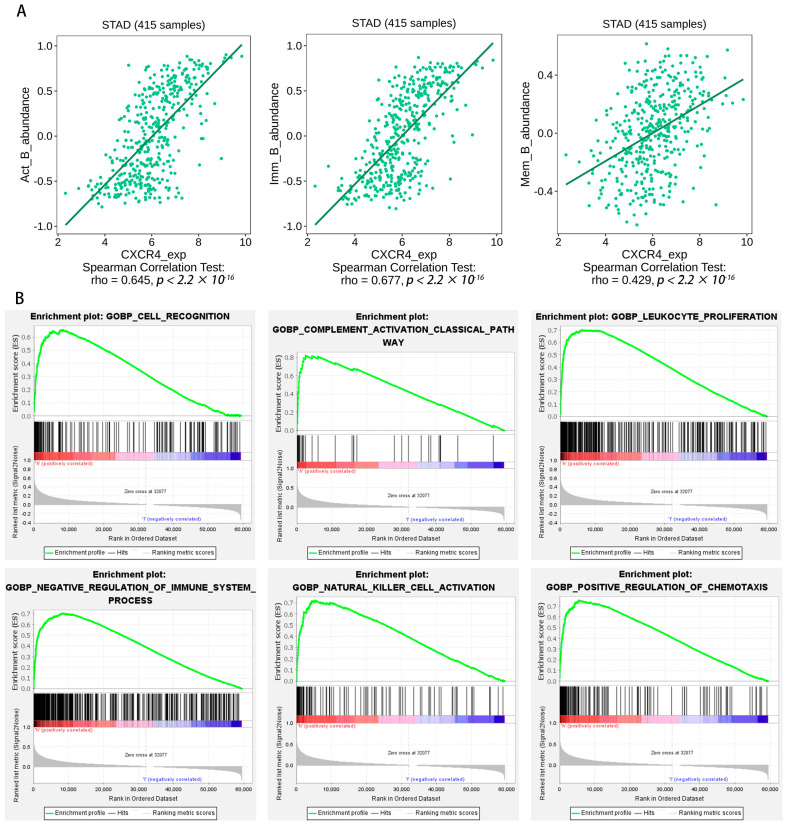
(**A**) Expression of *CXCR4* correlates with the number of activity, memory, and immature B cells. B-cell immune-related signatures may be found on the TISIDB website. (**B**) Gene set enrichment analysis. Exemplary enrichment plots created using GSEA indicate that the most enriched biological processes in the dataset, including *CXCR4*, are closely related to immune responses.

**Figure 8 ijms-24-12890-f008:**
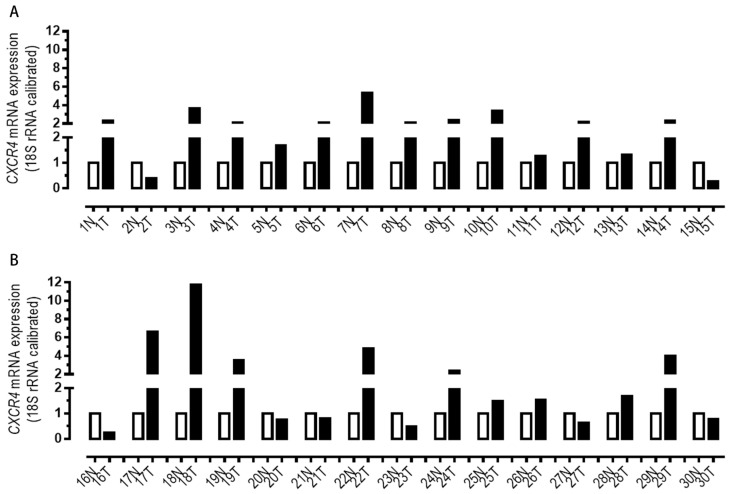
(**A**,**B**) RT-PCR analysis. The expression of CXCR4 between 30 pairs of normal and tumor tissues was examined via RT-PCR analysis, and CXCR4 levels were significantly elevated compared to normal tissue levels (22 out of 30). T, tumor; N, normal.

**Figure 9 ijms-24-12890-f009:**
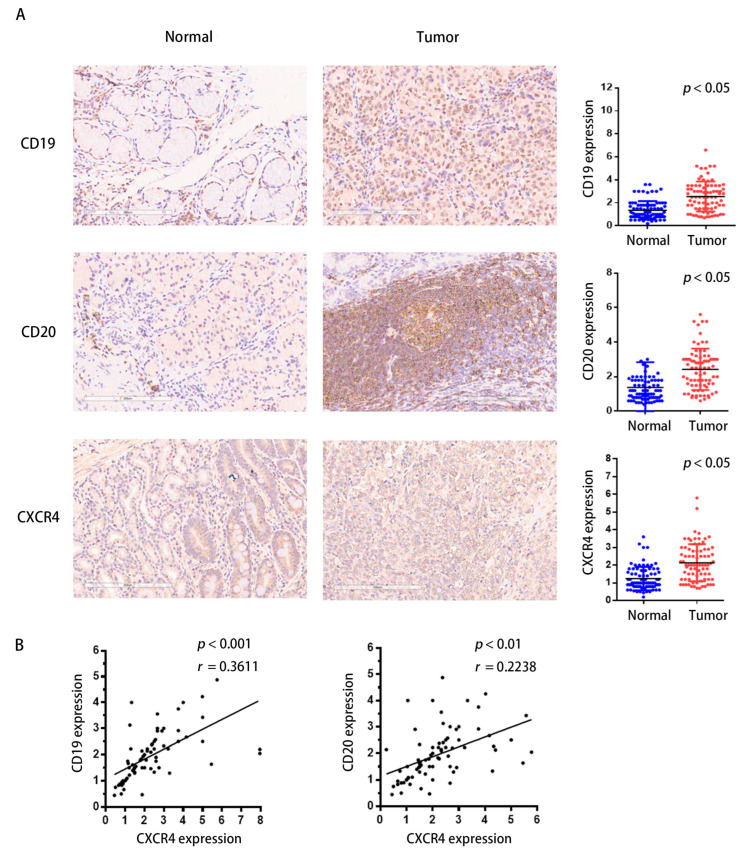
CXCR4 expression correlates positively with TIBs in clinical samples of STAD. (**A**) The immunohistochemical analysis. The slides of human gastric cancer tissues and paracancerous tissues were incubated with primary antibodies against CD20, CD19, and CXCR4 antibodies at 4 °C overnight. Afterward, the slides were incubated with a secondary antibody at room temperature for 30 min and stained with DAB substrate, followed by hematoxylin counterstaining. The immunohistochemical score was determined by multiplying the staining intensity score by the staining extent score. The color depth of brown was positively correlated with the protein expression level. (**B**) The expression levels of CXCR4, which is correlated with CD19 and CD20, were also evaluated. Statistically significant differences were determined at a *p*-value less than 0.05.

**Table 1 ijms-24-12890-t001:** Clinicopathological characteristics in STAD.

Clinicopathological Parameters	Expression		*p*-Value
	High	Low	
Age, years			
≤60	8	1	0.374
>60	14	7	
Gender			
Male	17	7	0.655
Female	5	1	
Location			
Cardiac	3	5	0.024
Antrum	6	1	
Pylorus	13	2	
Tumor size			
≤5 cm	16	4	0.384
>5 cm	6	4	
Pathology grade			
II	9	3	1
III	13	5	
TNM stage			
I + II	12	1	0.024
III + IV	10	9	

TNM, tumor node metastasis.

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
