# Peer review of "CXCR4 Expressed by Tumor-Infiltrating B Cells in Gastric Cancer Related to Survival in the Tumor Microenvironment: An Analysis Combining Single-Cell RNA Sequencing with Bulk RNA Sequencing"

_ijms, 2023, doi:10.3390/ijms241612890_

Round 1
Reviewer 1 Report
Dear Author,
The article titled "CXCR4 expressed by tumor-infiltrating B cells in gastric cancer 2 was survival related in the tumor microenvironment: analysis combining of scRNA-Seq with Bulk RNA-Seq" is well written and presented neatly.
Minor Comments:
I have a general question you mentioned after TISIDB database verification you concluded CXCR4 was a candidate gene for this study, why you have chosen GADD45B? We already know that in gastric cancer, CXC chemokines and chemokine receptors regulate the trafficking of cells in and out of the tumor microenvironment. CXC chemokines and their receptors can also directly influence tumorigenesis by modulating tumor transformation, survival, growth, invasion, and metastasis, as well as indirectly by regulating angiogenesis, and tumor-leukocyte interactions. GADD45B, an oncogene played a critical role in GC development. Moreover, GADD45B expression levels were closely tied to poor prognosis in GC patients, influencing the infiltration patterns of various cells within the tumor microenvironment, as well as impacting the metabolic pathways involved in GC progression. Did the authors have any chance to correlate GADD45B with CD19 and CD20 expression, if the authors have studied both the CXCR4 and GADD45B it would be interesting to see the outcome.
On page 6, line 139 you mentioned through λ through cross-validation you have shortlisted three important genes but you have mentioned only 2 genes (CXCR4 & GADD45B) what is the other shortlisted gene, kindly add that too.
Author Response
Dear Editor and reviewer,
On behalf of my co-authors, we thank you very much for giving us an opportunity to revise our manuscript, we appreciate the editor and the reviewers very much for their positive and constructive comments and suggestions on our manuscript entitled “CXCR4 expressed by tumor-infiltrating B cells in gastric cancer was survival related in tumor microenvironment: analysis combining of scRNA-Seq with Bulk RNA-Seq” (Manuscript ID: ijms-2527944) We have studied the editor’s and the reviewers’ comments carefully and have tried our best to revise our manuscript according to the comments. We have adopted all of the suggestions. With your remarkable help, we have made much revision, we think that the manuscript has been greatly improved by these revisions and we hope that you will now find it suitable for publication in the INTERNATIONAL JOURNAL OF MOLECULAR SCIENCES. Our point-by-point responses to comments are detailed in attachment.
We are looking forward to hearing from you at your earliest convenience.
Yours sincerely,
Jingjing Hou, Ph.D.
Zhongshan Hospital of Xiamen University
201-209 South Hubin Rd., Xiamen, Fujian province 361004, China
Tel: +86 592 2292799, Fax: +86 592 2297189
Email:[email protected]

Reviewer 2 Report
In this study, to better understand the role of tumor-infiltrating B cells (TIBs) in Gastric Cancer (GC), which is one of the leading cancers, this study used scRNA-Seq and bulk RNA-Seq data to identify candidate hub genes. The study used a combination of scRNA-Seq and bulk RNA-Seq data, which allowed for a more comprehensive analysis of the cellular composition of STAD tumors. The study identified CXCR4 as a hub gene for TIBs in GC. A total of 15 cell clusters were classified in the scRNA-Seq database. The green module, which is most associated with cancer and B cells, was identified using WGCNA. The intersection of 13 genes in two separate datasets (scRNA, Bulk) were attained for further analysis. Survival studies revealed that increased CXCR4 expression was linked to worse overall survival. CXCR4 expression was correlated with the activity of both memory and immature B cells in STAD. CxCR4 overexpression which was further confirmed by IHC and RT-PCR. This suggests that CXCR4 may play a critical role in the recruitment and activation of TIBs in tumors. The study found that increased CXCR4 expression was linked to worse overall survival in patients with GC suggesting that CXCR4 may be a potential therapeutic target for GC. This study adds promising information which would help to improve gastric cancer patient treatment management.
The article flows well and is scientifically relevant. It is presented in a comprehensive manner, but poorly written, with only minor concerns to be taken care of:
1. Use bigger fonts in almost all the figures and keep the font size while labelling axes consistent across board. The use of tense throughout is inconsistent-present and past, active, and passive. Keep it consistent.
2. Line 43, …the fourth largest leading..” change it to-.. is the fourth leading cause of ..
3. Line 44: Start the statement with: Despite advancements in treatment, such as molecularly targeted therapy, chemotherapy, and surgery, GC outcomes remain dismal.
4. Figure 2, a, b, c- increase the font size, elaborate the colors in the legend as well. D- Label axes (include x & y axes names) and what do the scatter and the peak show. E- define category (colors) in the legend, increase font size to be readable by reader.
5. Line 73 Expand FC-what does FC stands for. Mention once and use FC thereafter.
6. Line 78 Rephrase it to--Marker genes for B cells were found to be those that were uniquely expressed.” Line 79 Rephrase as--Figure 2E displays the GO analysis results, which showed that the marker genes of the B cells …
7. Result 2.2- Please discuss the result shown in the heat map of figure 3. Elaborate on the data received and the invading pattern.
8. Line 111-what does this line mean-- . Use a heat map to examine the relationship between the many types of cancer, B cells, 111 and the module's most important genes.
9. Does the author mean to say—“we used a heat map to examine the relationship between the different types of cancer, B cells, and the module's most important genes.
10. Line 106 Add superscript 2 with R
11. Line 113 Rephrase it as - This suggests that the top 2,000 genes …
12. Figure 4 shows A, B and C while in the legend it is mentioned A, B, C, and D. Please correct the labelling in plot shown in the figure. 4A-left: y axis has mis printed/labled fonts, correct it in bigger font. Please elaborate about the plots in the legend like this plot shows –(left) and ..(right). In A & B the font size is so small, very difficult to read while in C it is bigger. Please keep the font size consistent, bigger across. I am assuming D corresponds to the C in the figure. Rewrite the figure legend and include more information shown in the plots and cluster dendrogram.
13. Line 124 Title 2.4-change it to- Analysis of the relationship between key genes and prognosis in gastric cancer.
14. Line 125-rephrase We gained an intersection…to “we identified…”. Add more information for example- discuss the result which were those genes found to be associated with prognosis-highlight them.
15. Figure 5- Add elaborated legend.
16. Figure 6- Increase the font size, it is impossible to read A, B and C. Add elaborate figure legend explaining what is shown in the graph, not just the title.
17. Figure 8. Please include in the legend what does the two graphs signify, in what way they are different. Mark them as A and B as well. Include N and T in the legend where you have mentioned normal and tumor.
18. Figure 9. Include a title line explaining the figure before A. Please elaborate which tissue/ or part of tissue is shown in the figure. What is the stain and counterstain used, and what does it signify and why. Plots shown in 9B are pasted as picture together in poor quality. Please paste them individually, with bigger fonts, P value shown above the plot, not on the scatter plot. Label images and each plots as well and include an appropriate legend.
19. Result 2.10 , elaborate a bit more the findings shown in Fig 9.
English language needs to be corrected at a few places as mentioned.
Author Response

(The authors gave the same response as above.)

Reviewer 3 Report
This study is an interesting platform which used scRNA-Seq and bulk RNA-Seq data to identify candidate hub genes in gastric cancer (GC). The study found that CXCR4 is highly expressed by tumor-infiltrating B cells (TIBs) in GC, suggesting it may serve as a hub gene for these cells. The findings suggest that increased CXCR4 expression is linked to worse overall survival and may serve as a starting point for future research into the molecular mechanisms by which immune cells gain access to tumors and potentially identify therapeutic targets.
This study is extremely interesting, but it needs major language check as it was extremely difficult to connect the dots.
Author Response
Dear reviewer,
On behalf of my co-authors, we thank you very much for giving us an opportunity to revise our manuscript, we appreciate the editor and the reviewers very much for their positive and constructive comments and suggestions on our manuscript entitled “CXCR4 expressed by tumor-infiltrating B cells in gastric cancer was survival related in tumor microenvironment: analysis combining of scRNA-Seq with Bulk RNA-Seq” (Manuscript ID: ijms-2527944) We have studied the editor’s and the reviewers’ comments carefully and have tried our best to revise our manuscript according to the comments. We have adopted all of the suggestions. With your remarkable help, we have made much revision, we think that the manuscript has been greatly improved by these revisions and we hope that you will now find it suitable for publication in the INTERNATIONAL JOURNAL OF MOLECULAR SCIENCES. Our point-by-point responses to comments are detailed in the attachment.
We are looking forward to hearing from you at your earliest convenience.
Yours sincerely,
Jingjing Hou, Ph.D.
Zhongshan Hospital of Xiamen University
201-209 South Hubin Rd., Xiamen, Fujian province 361004, China
Tel: +86 592 2292799, Fax: +86 592 2297189
Email:[email protected]
